# Disrupted hemodynamic response within dorsolateral prefrontal cortex during cognitive tasks among people with multiple sclerosis-related fatigue

**Bruna D. Baldasso[1], Syed Z. Raza[1], Sadman S. Islam[1,2], Isabella B. Burry[1], Caitlin J. Newell[1], Sydney R. Hillier[1], Michelle Ploughman[1]** *

1 Recovery & Performance Laboratory, Faculty of Medicine, Memorial University of Newfoundland, St. John's, NL, Canada, 2 Computer Science, Faculty of Science, Memorial University of Newfoundland, St. John's, NL, Canada

* michelle.ploughman@med.mun.ca

## Abstract

### Introduction

Mental fatigue is an early and enduring symptom in persons with autoimmune disease particularly multiple sclerosis (MS). Neuromodulation has emerged as a potential treatment although optimal cortical targets have yet to be determined. We aimed to examine cortical hemodynamic responses within bilateral dorsolateral prefrontal cortex (dlPFC) and fronto-polar areas during single and dual cognitive tasks in persons with MS-related fatigue compared to matched controls.

### Methods

We recruited persons (15 MS and 12 age- and sex-matched controls) who did not have physical or cognitive impairment and were free from depressive symptoms. Functional near infrared spectroscopy (fNIRS) registered hemodynamic responses during the tasks. We calculated oxyhemoglobin peak, time-to-peak, coherence between channels (a potential marker of neurovascular coupling) and functional connectivity (z-score).

### Results

In MS, dlPFC demonstrated disrupted hemodynamic coherence during both single and dual tasks, as evidenced by non-significant and negative correlations between fNIRS channels. In MS, reduced coherence occurred in left dorsolateral PFC during the single task but occurred bilaterally as the task became more challenging. Functional connectivity was lower during dual compared to single tasks in the right dorsolateral PFC in both groups. Lower z-score was related to greater feelings of fatigue. Peak and time-to-peak hemodynamic response did not differ between groups or tasks.

**Data Availability Statement:** All data are in the paper and its supporting information files.

**Funding:** The research was funded by Canada Research Chairs (Grant Number 2019-00290) and Canada Foundation for Innovation (Grant Number 33621) (MP). The funders were not involved in the conduct of the research.

## Conclusions

Hemodynamic responses were inconsistent and disrupted in people with MS experiencing mental fatigue, which worsened as the task became more challenging. Our findings point to dlPFC, but not frontopolar areas, as a potential target for neuromodulation to treat cognitive fatigue.

## Introduction

Mental fatigue is experienced by almost everyone at one time or another, especially among workers who are sleep-deprived [1] and those whose work involves sustained attention such as pilots [2] and long haul drivers [3, 4]. Many persons with traumatic brain injury [5], systemic lupus erythematosus [6], inflammatory bowel disease [7] and multiple sclerosis (MS) [8] feel mental fatigue unceasingly which disrupts work productivity and quality of life.

Several groups are testing non-invasive brain stimulation to treat fatigue [9, 10], especially in MS [11–13]; a method that excites or inhibits neuronal networks on the cortical surface. In studies of brain stimulation for MS-related fatigue, cortical targets include the right parietal [13, 14], right frontal [13], and left dorsolateral prefrontal [12, 14, 15] areas, with varying degrees of treatment efficacy. Identifying the appropriate targets for stimulation is an emerging field in fatigue research. Using functional magnetic resonance imaging, studies confirm that functional network connectivity, more than structural damage, underlies MS-related fatigue [11, 16] especially in prefrontal [17] and sensorimotor [18] areas. Functional near-infrared spectroscopy (fNIRS) is a relatively new technique to probe task-induced changes to blood oxygenation, a proxy of neuronal activation, on the cortical surface [19]; the target of brain stimulation therapies for fatigue. Compared to functional magnetic resonance imaging, fNIRS is inexpensive, less sensitive to movement and has fewer contraindications (e.g. metal implants) [20]. For instance, using fNIRS, Skau and colleagues reported reduced activity in frontopolar and dorsolateral prefrontal cortices (PFC) during a mentally fatiguing task among people with traumatic brain injury [21].

Using fNIRS in people with MS, both Broscheid et al. [22] and Saleh et al. [23] implicated premotor cortices during a dual task condition that involved walking while performing a mentally fatiguing cognitive task. Compared to single tasks, during dual tasks there was greater recruitment of right premotor cortex [22]. Although providing the first evidence of potential target areas related to mental fatigue using fNIRS, both studies tested combined motor and cognitive tasks so it is difficult to discern key cortical regions of interest to treat mental fatigue specifically. Only one study [24] has examined brain oxygenation during high and low load cognitive tasks using fNIRS among people with MS. The results were inconclusive, showing no differences between regions (ventrolateral PFC, dorsolateral PFC and inferior parietal) and between low and high load cognitive tasks in MS and healthy controls. However, lower oxygenation in dorsolateral PFC was significantly associated with greater subjective cognitive fatigue in the MS group. Group differences in age (younger controls) and cognitive impairment at baseline could have influenced the results. It is important in future research to ensure groups are matched and that there are no other confounding conditions, particularly among the MS participants, such as cognitive impairment, physical disability or depression, which could affect capacity for brain response [25–29].

Here, we studied mental fatigue during single and dual cognitive tasks using fNIRS. We compared age and sex-matched controls to a group of people with MS who reported fatigue but

did not have depression, physical or cognitive impairments. We evaluated four different features of the hemodynamic response: peak oxy-hemoglobin concentration, time-to-peak concentration, correlations between fNIRS channels (intra-regional coherence) and functional connectivity (z-score) in bilateral dorsolateral PFC (Brodmann area 9/46) and frontopolar cortex (Brodmann area 10). We hypothesized that participants with MS would demonstrate decreased activation of dorsolateral PFC compared to controls, which would worsen with more complex cognitive tasks. We expected that fNIRS indices would be related to feelings of fatigue.

## Methods

### Participants

Following approval by the local Health Research Ethics Board (HREB#2021.005) according to the Declaration of Helsinki, we approached patients attending a specialized MS neurology clinic and through poster advertisement. After obtaining informed written consent, we determined whether they were eligible for the study from interview and review of health records. They were included if they, 1) were diagnosed with relapsing-remitting MS by a neurologist according to the McDonald criteria [30], 2) relapse-free during the previous three months, 3) 18 years or older and 4) Expanded Disability Status Scale (EDSS) score ≤3.0 indicating they did not have mobility impairment. Participants were excluded if, 1) they had been diagnosed with neurological conditions other than MS, 2) they had untreated cardiovascular disease, which could potentially affect brain hemodynamics, or 3) they scored lower than 26 on the Montreal Cognitive Assessment, which is considered abnormal [31], 4) they reported moderate depressive symptoms, scoring more than 10 out of 21 on the depression subscale of the Hospital Anxiety and Depression Scale [32], or 4) they were pregnant. We recruited healthy controls that were matched to the MS participants for age (±3 years) and sex using posters, social media and word-of-mouth. We used the same inclusion and exclusion criteria with the exception of the MS-specific criteria. Control participants were matched to more than one MS participant. Data was collected between May 21 and September 28, 2021.

Based on previous research, we aimed to recruit at least 15 persons with MS. We did not complete sample size calculation because of inconclusive findings from previous work [22, 23] and lack of a primary outcome (considering four fNIRS variables; peak oxy-hemoglobin concentration, time-to-peak concentration, intra-regional correlations and functional connectivity).

### Baseline measures

Participants were asked to refrain from caffeine, smoking and using other stimulants at least 24h before the experimental procedure, which was completed at a single visit. They were also asked to follow their usual sleep schedule, to prevent alterations in circadian rhythm. Participants provided demographic and health information, including age, years of education, year of MS diagnosis, type of MS, comorbid conditions, and list of medications. They completed the Hospital Anxiety and Depression Scale (HADS), which consisted of 14 questions about depression and anxiety symptoms during the previous two weeks. We considered only the depression subscale score. Each question ranged from 0 to 3 points, adding to a maximum of 21 points for each subscale [32]. The Montreal Cognitive Assessment measured overall cognitive function across five domains: visuospatial and executive function, attention, language, memory, and orientation. Scores on each domain summed to a maximum of 30 points. Scores lower than 26 are considered abnormal [31]. Participants completed the Symbol-Digit Modality Test to assess information processing speed. They were asked to match symbols and

numbers, following an answer key. Scores were calculated as the number of correct matchings in 90 seconds [33].

## Fatigue measures

Two different subjective measures characterized participants' feelings of mental/cognitive fatigue: 1) subjective cognitive fatigability related to the tasks ('state' fatigue [25, 29]) and 2) prolonged disease-related 'trait' fatigue. State fatigue was measured using a visual analogue scale. Participants were asked to rate their present experience of fatigue by drawing a vertical line on a 100 mm scale ranging from 0 (no fatigue) to 100 (extreme fatigue). The distance between 0 and the vertical line provided the score. Trait fatigue was measured using the Fatigue Scale for Motor and Cognitive Functions (FSMC). Participants completed a 20-item questionnaire assessing fatigue in their everyday life with a five-point Likert scale (0 = Does not apply at all, 5 = Applies completely). The tool has been validated for use in healthy and MS populations [34]. We considered only the score on the cognitive subscale, which ranges from a low of 0 to a high of 50.

## fNIRS system and optode array

fNIRS data was collected using the continuous-wave NIRScoutX® 16x16 imaging system (NIRx Medical Technologies, Berlin, Germany) containing 16 LED sources and 16 detectors, at a sampling rate of 3.9 Hz. For the purpose of this study, eight sources and eight detectors were used for analysis, with a total of 14 channels covering the prefrontal cortex. Probe arrangement on the cap was determined using the fNIRS Optode Location Decider (fOLD) software (54) with sensitivity ranging from 43.1 to 87.4. The final montage covered the dorso-lateral and frontopolar areas (Brodmann areas 9/46 and 10, respectively) in both hemispheres (Fig 1). Distance between sources and detectors was 3 cm, kept constant by spacers. A bundle of short-distance detectors, attached to the optodes, measured and removed data regarding superficial scalp blood flow. Individual cap sizes were determined prior to testing by

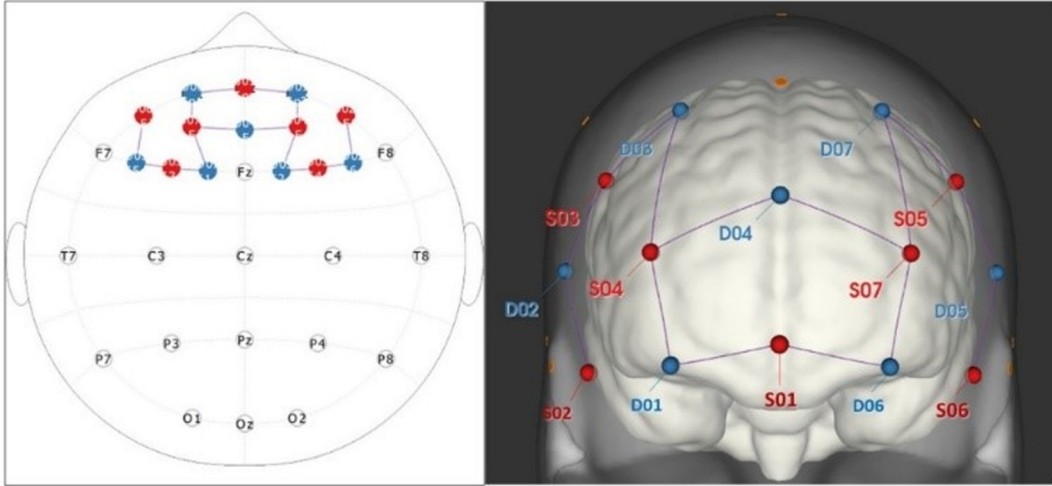

**Fig 1. fNIRS channel montage.** Configuration of channels in the fNIRS cap covering four regions of interest (ROI): right dorsolateral prefrontal cortex (S02-D02, S03-D03, S03-D02, S04-D03); left dorsolateral prefrontal cortex (S05-D05, S05-D07, S06-D05, S07-D07); right frontopolar area (S01-D01, S04-D01, S04-D04); left frontopolar area (S01-D06, S07-D04, S07-D06). Each line represents a channel with a source and a detector; seven in each hemisphere. Red labels represent sources (S) and blue labels represent detectors (D).

measuring head circumference aligned with the nasion and the inion. Prior to data recording, probes were calibrated, and each channel was inspected for signal quality and noise. A signal was considered acceptable if the gain was equal or greater than seven, and if noise level was less than 7.5. All data recordings were completed using the NIRStar® acquisition software (NIRx Medical Technologies, Berlin, Germany).

## Procedure

Participants sat comfortably facing a screen approximately 2.5m away. We provided standardized instructions followed by a practice trial without wearing the fNIRS cap. The cap was then fitted to the head, and proper probe placement was ensured by using the inion as a reference point. To prevent excessive environmental light, black-out curtains were shut and the lights turned off.

The experiment involved completing two types of cognitive tasks, a single cognitive task and a dual cognitive task, presented randomly. After hearing an auditory signal, the single task involved serially subtracting 7's, starting at a random number presented on the screen. Participants continued to subtract for 20s, providing their answers audibly, until the stop signal indicated a rest period (20s). The dual-task involved serially subtracting 7's, starting at a random number in the same way as the single cognitive task except participants alternated their answers with letters of the alphabet in ascending order. Stimulus presentation followed a randomized block design, in which a block corresponded to 20s of rest and 20s of task (Fig 2), repeated for 10 blocks. Instructions for each task appeared on the screen 6 seconds in advance of the task. Stimulus presentation was built and randomized using the NIRStim® platform (NIRx Medical technologies, Berlin, Germany).

## Data analysis

**Preprocessing of fNIRS data.** Data was pre-processed using Homer3® (55), an application based in MATLAB® version 9.3.0 R2017b (The Mathworks Inc., Massachusetts, USA). A processing pipeline was built for data filtering, noise removal, and transformation of light intensity to hemoglobin concentrations. First, channels with a signal to noise ratio lower than 8 were

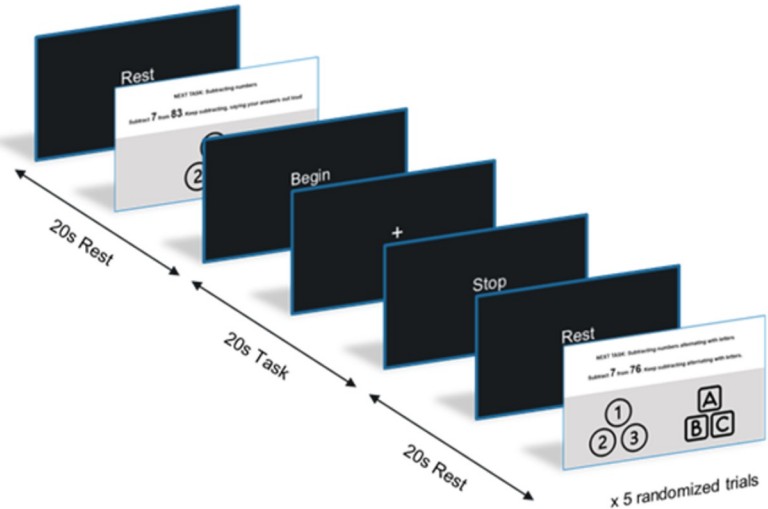

**Fig 2. Task sequence arranged in a block design.** Participants executed the task for 20s, followed by 20s of rest. Task order was randomized for each participant.

excluded (hMR_prune_and light intensities were converted to optical density (hmR_Intensity2OD). Next, we identified motion artifacts using a standard deviation threshold of 20. Spikes with SD > 20 within an interval of 5s were labeled as artifacts. Motion artifacts were corrected using the Spline and Wavelet methods combined [35]. Next, a high pass filter of 0.01 Hz and a lowpass filter of 0.09 Hz was added to remove cardiac and respiration signals. Optical density was then converted to oxy (HbO), deoxy (HbR), and total (HbT) hemoglobin concentrations using the modified Beer-Lambert Law. Subject-level analysis was completed by applying a general linear model (GLM), using the method of least squares and a Gaussian function. Short separation channels (source-detector distance less than 15 mm) were regressed out to remove information regarding superficial scalp blood flow. The hemodynamic response time series for each subject was extracted for further analysis. HbO and de-oxyhemoglobin (HbR) values for all brain regions of interest for each participant can be found in the S1 dataset.

**Statistical analysis.** We examined baseline differences in demographics and state and trait fatigue between groups using ANOVA. We determined changes in state fatigue before and after the experiment using one-sided paired samples t-tests. We evaluated four different features of the hemodynamic response: peak HbO, time-to-peak, intra-regional correlations (between-channel coherence), and functional connectivity (z-score) in bilateral dorsolateral PFC and frontopolar areas (four regions-of-interest). We first calculated both peak HbO concentration and time-to-peak. Peak HbO concentrations were identified by taking the maximum or minimum absolute value between 7-17s for each channel. Time-to-peak was obtained by taking the time point corresponding to the peak HbO, in seconds. For intra-regional (between-channel) analyses, we correlated HbO concentrations, within the window of 5 to 20s for each channel for each group in single and dual task conditions. In order to display the direction of the R-value, R-values in the positive direction (R = 1.0) were represented as red and R-values in the negative direction (R = -1.0) represented in blue on a heat map. We calculated functional connectivity at subject-level for channels in each region-of-interest using pairwise Pearson's correlations (HbO concentrations within the window of 5 to 20s between time series). Pearson's R coefficients were then normalized using the Fisher-z transformation, and averaged for each region-of-interest, for each participant (z-score). Because positive and negative correlations cancel each other out, we removed the sign of the coefficients to reflect the strength of the connectivity, rather than direction. Higher z-scores indicated greater connectivity. Peak HbO, time-to-peak and functional connectivity scores were then compared between groups and conditions (single and dual task) using a factorial ANOVA (group X condition), reporting the variation between sample means as F-value. We calculated the strength of the relationships between trait fatigue (FSMC-Cognitive subscale) and peak HbO, time-to-peak and connectivity z-score using Pearson's correlations. Correlations <0.30 were considered weak, 0.3 to 0.7 moderate and more than 0.70 considered strong. P-values were significant at $p \leq 0.05$ and analysis was carried out in SPSS® (IBM version 28).

## Results

Fifteen people with MS (12 females) and 12 healthy controls (10 females) matched for age and sex consented to participate in the study. MS patients had mild or no disability, with median Expanded Disability Status Scale (EDSS) score of 0, ranging from 0–3. MS disease duration was 15.47 years (6.45 SD). There were no significant differences between groups in age, years of education, depressive symptoms, or processing speed (SDMT; Table 1). Despite screening for cognitive impairment using the MoCA, control participants scored significantly higher than MS (p = 0.034).

**Table 1. Participant characteristics.**

| | MS (N = 15) | | Controls (N = 12) | | | |
|---|---|---|---|---|---|---|
| | **Mean** | **SD** | **Mean** | **SD** | **F** | **P** |
| **Age** | 43.80 | 8.87 | 45.25 | 10.52 | 0.151 | 0.701 |
| **Years of Education** | 15.73 | 2.58 | 19.17 | 6.03 | 3.988 | 0.057 |
| **Depression (HADS)** | 3.47 | 2.48 | 2.25 | 3.31 | 1.198 | 0.284 |
| **Cognition (MoCA)** | 26.93 | 1.44 | 28.08 | 1.17 | 5.027 | **0.034*** |
| **Processing speed (SDMT)** | 54.67 | 11.69 | 59.08 | 7.66 | 1.271 | 0.270 |
| **State Fatigue (VAS)** | 31.27 | 28.64 | 19.00 | 23.48 | 1.429 | 0.243 |
| **Trait Fatigue (FSMC)** | 31.33 | 12.33 | 18.08 | 7.87 | 10.412 | **0.003*** |

MS: multiple sclerosis, HADS: Hospital Anxiety and Depression Scale, MoCA: Montreal Cognitive Assessment, SDMT: Symbol Digit Modalities Test, VAS: Visual Analogue Scale, FSMC: Fatigue Scale for Motor and Cognitive Functions (Cognitive subscale).

* significantly different p<0.05.

As expected, the MS group scored significantly higher in trait fatigue than controls (Table 1 and Fig 3A). Although both groups had similar levels of state mental fatigue at baseline, measured using the visual analogue scale, only the MS group showed a significant increase in task-induced fatigue (controls, t = -0.833, p = 0.214; MS, t = -3.520, p = 0.002; Fig 3B).

## Hemodynamic response

Time series for each channel were plotted (Fig 4A–4D). Visual inspection of the plots suggested that there was a task-related decrease in HbO among controls (Fig 4A and 4B) and asynchrony across channels in the MS group (Fig 4C and 4D) during both single and dual tasks. However, when considering all channels, comparing MS and controls, and single and dual tasks, there were no significant differences in average peak HbO or time-to-peak (p>0.05 See S1 Dataset). There were also no significant differences in peak HbO or time-to-peak between MS and controls or between tasks in any of the four subregions assessed (Table 2).

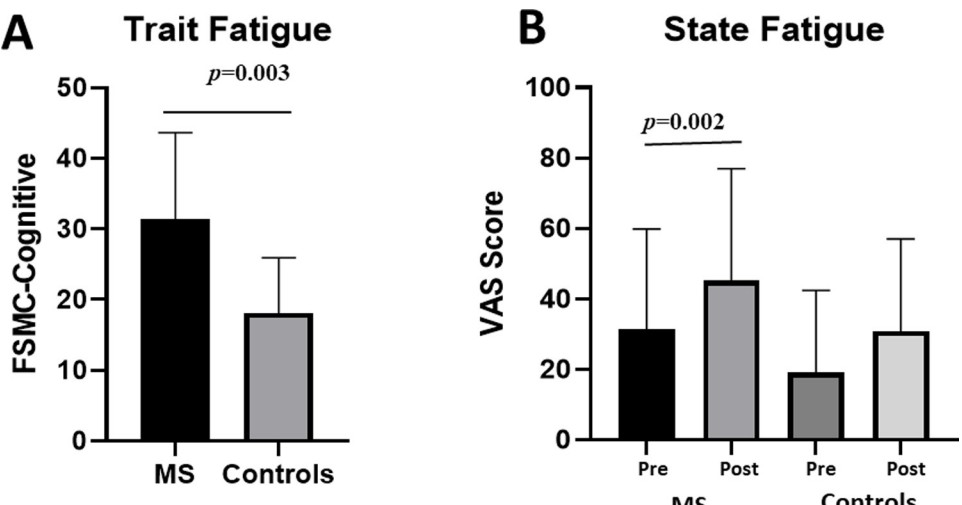

**Fig 3. Differences in trait and state fatigue between MS and controls.** (A) Trait fatigue was significantly higher in MS compared to controls. (B) MS participants showed a significantly higher state fatigue after the experiment compared to baseline. Although non-significant, levels of fatigue in controls increased slightly after the procedure. VAS: Visual Analogue Scale, FSMC: Fatigue Scale for Motor and Cognitive Functions.

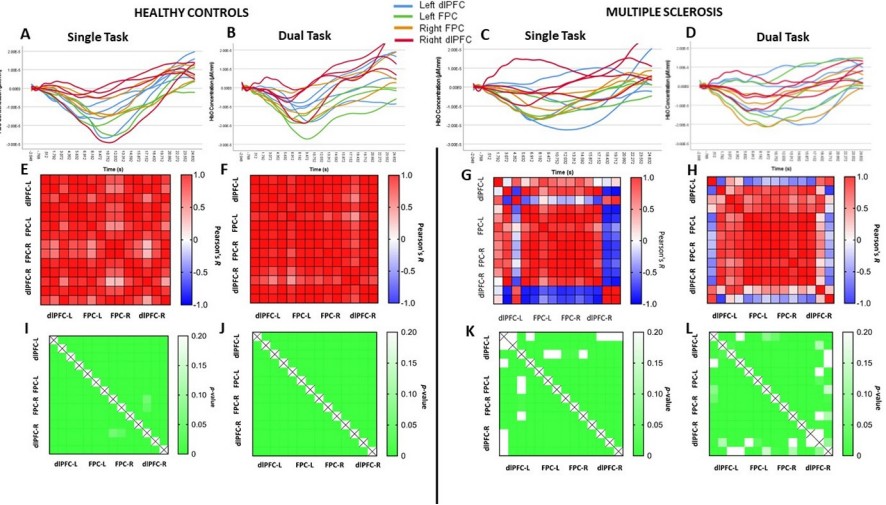

**Fig 4. Oxyhemoglobin (HbO) levels detected in channels in four prefrontal cortex subregions.** Plots of left dorsolateral prefrontal cortex (dlPFC, blue), left frontopolar cortex (FPC, green), right frontopolar cortex (FPC, orange) and right dorsolateral prefrontal cortex (dlPFC, red) in controls (**AB**) and multiple sclerosis (MS) (**CD**) during single (**AC**) and dual (**BD**) tasks. Correlograms of channels in prefrontal cortex in controls (**EF**) and MS (**GH**) during single (**EG**) and dual (**FH**) tasks. Each block shows the relationship between two channels (R-values); red represents positive correlations and blue represents negative correlations. Bottom panels are heat maps representing correlation *p*-values in controls (**IJ**) and MS (**KL**) during single (**IK**) and dual (**JL**) tasks. Lower *p*-values are brighter green. R, right; L, left; dlPFC, dorsolateral prefrontal cortex; FPA, frontopolar area.

**Table 2. Group and task differences in peak oxy-hemoglobin concentration, time-to-peak concentration, and functional connectivity (z-score) by region of interest.**

| Effects by Region of Interest | Peak HbO | | Time-to-Peak | | Functional Connectivity | |
|---|---|---|---|---|---|---|
| | *F* | *p* | *F* | *p* | *F* | *p* |
| **Left dlPFC** | | | | | | |
| Group | 0.054 | 0.819 | 0.144 | 0.707 | 3.925 | 0.059 |
| Task | 2.028 | 0.167 | 0.547 | 0.466 | 0.008 | 0.930 |
| Group*Task | 0.124 | 0.727 | 2.616 | 0.118 | 0.898 | 0.352 |
| **Right dlPFC** | | | | | | |
| Group | 0.005 | 0.945 | 0.819 | 0.374 | 2.189 | 0.151 |
| Task | 0.045 | 0.833 | 0.609 | 0.442 | 5.549 | 0.027* |
| Group*Task | 2.102 | 0.160 | 1.239 | 0.276 | 2.502 | 0.126 |
| **Left FPA** | | | | | | |
| Group | 0.698 | 0.411 | 0.324 | 0.575 | 0.556 | 0.463 |
| Task | 0.7 | 0.411 | 0.048 | 0.828 | 1.24 | 0.276 |
| Group*Task | 0.466 | 0.501 | 0.714 | 0.406 | 0.505 | 0.484 |
| **Right FPA** | | | | | | |
| Group | <0.001 | 0.988 | 0.993 | 0.329 | 0.009 | 0.924 |
| Task | 0.456 | 0.506 | 1.439 | 0.242 | 0.051 | 0.823 |
| Group*Task | 0.781 | 0.385 | 0.001 | 0.981 | 0.982 | 0.332 |

Notes: dlPFC: dorsolateral prefrontal cortex. FPA: frontopolar area. HbO: oxygenated hemoglobin.

*significant p<0.05.

## Coherence of hemodynamic response

To visualize intra-regional coherence between channels, we built correlograms with R coefficients (Fig 4E–4H) and significance values for each pair of channels (Fig 4I–4L). While controls showed a more consistent and stronger coherence between channels in all prefrontal regions (statistically significant and strong R-values; Fig 4E and 4F/4I and 4J), we observed low coherence in the MS group, with a larger amount of weak and non-significant correlations for both conditions (Fig 4G and 4H/4K and 4L). Specifically, the disrupted coherence appeared to be concentrated in the bilateral dorsolateral PFC rather than the frontopolar areas. Low coherence was most prominent in left dorsolateral PFC during the single task (Fig 5B, 5E and 5F) but occurred bilaterally during the more challenging dual task in the MS group (Fig 5G–5J).

## Functional connectivity

When considering all channels, there were no differences in functional connectivity z-scores between MS and controls however there was significantly greater functional connectivity during single (1.30, SD 0.39) compared to dual tasks (1.14, SD 0.31, $t$ = 2.066, $p$ = 0.049). Within regions of interest, a significant effect of task was found with greater connectivity in right dorsolateral PFC during single compared to dual tasks (Table 2). The effect of group approached significance in the left dorsolateral PFC (Table 2) with somewhat greater connectivity in controls compared to MS participants. There were no significant effects of group or task on functional connectivity in left or right frontopolar areas.

## Relationships between subjective fatigue and connectivity

When considering all channels in both hemispheres in both groups, during the single task, lower connectivity (z-score) was related to greater feelings of fatigue during everyday life on the cognitive subscale of the FSMC (R = -0.386, p = 0.047; Fig 6). There were no other

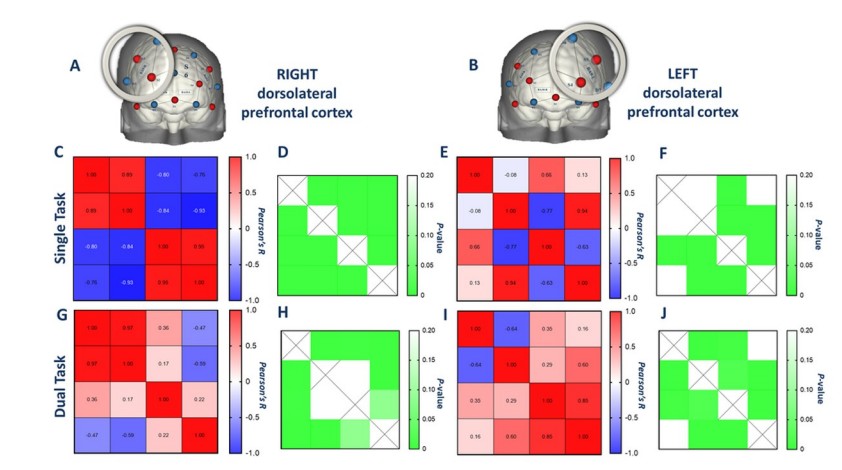

**Fig 5. Oxyhemoglobin (HbO) levels detected in channels in right and left dorsolateral prefrontal cortices in the multiple sclerosis (MS) group. A**. Right and **B.** Left dorsolateral prefrontal cortex correlograms showing coherence between channels during single and dual tasks (**CEGI**). Each block shows the relationship between two channels (R-values); red represents positive correlations and blue represents negative correlations. Panels to the right of correlograms are heat maps representing correlation *p*-values during single and dual tasks (**DFHJ**). Lower *p*-values are brighter green. R, right; L, left; dlPFC, dorsolateral prefrontal cortex; FPA, frontopolar area.

significant relationships between HbO concentration, time-to-peak, connectivity and feelings of fatigue (p>0.05, See S1 Dataset).

## Discussion

We aimed to determine cortical regions implicated in mentally fatiguing cognitive tasks in order to identify target areas for brain stimulation to treat fatigue, especially in MS. As expected, persons with MS reported greater cognitive fatigue and felt greater cognitive fatigability during the cognitive tasks. We report three main findings. First, in MS participants, the right and left dorsolateral PFC demonstrated reduced coherence of hemodynamic response during both single and dual cognitive tasks, as evidenced by non-significant and negative correlations between fNIRS channels positioned in those regions (Fig 4). In MS, reduced coherence occurred in left dorsolateral PFC during the single cognitive task but occurred bilaterally as the task became more challenging (dual task: Fig 5). Secondly, when calculated as a z-score within regions, functional connectivity was lower during dual compared to single tasks in the right dorsolateral PFC in both groups (Table 2). Lower functional connectivity was related to greater feelings of fatigue in everyday life in both MS and controls (Fig 6). Finally, there were no significant differences between groups or tasks in peak and time-to-peak task-induced oxyhemoglobin levels.

### Prefrontal cortex as a potential target for neuromodulation for mental fatigue

The dlPFC (Brodmann's areas 9 and 46), a densely interconnected region important for working memory and motor planning, receives input from and sends projections to numerous cortical and subcortical structures (e.g. limbic and temporal cortices, reticular formation, caudate, substantia nigra, thalamus) [36]. Lesions in dlPFC disrupt monitoring of information during verbal and spatial working memory tasks [37], while sparing performance on simple choice-based memory tasks [36]. The relationships between mental/cognitive fatigue and dlPFC is less clear. Using fNIRS during a verbal fluency task, Suda and group [38] reported that in

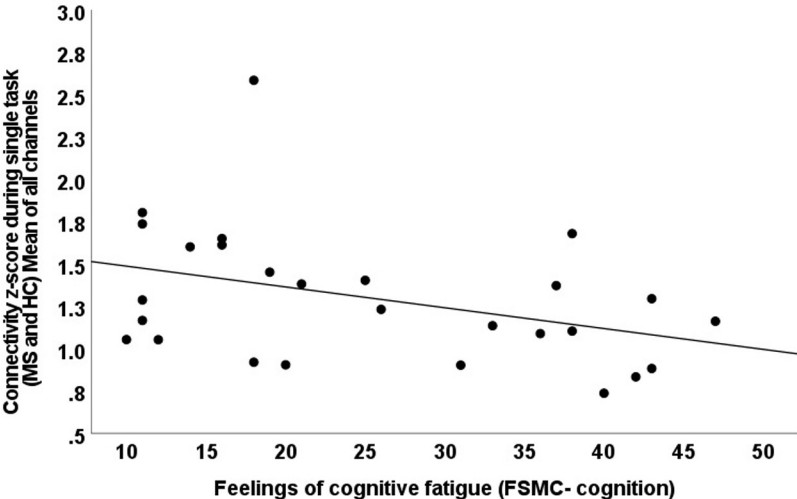

**Fig 6. Relationship between feelings of cognitive fatigue, in multiple sclerosis and controls, and functional connectivity (z-score).** Z-score is average of all channels for both groups (R = -0.386, p = 0.047). MS, multiple sclerosis; FSMC, Fatigue Scale for Motor and Cognitive functions.

healthy subjects, greater subjective fatigue was related to lower oxy-hemoglobin levels in bilateral ventrolateral PFC, while shorter sleep duration was related to lower oxy-hemoglobin levels in bilateral dlPFC [39]. In examining the pathological correlates of MS-related fatigue, Jaeger and group [39] showed that among persons with MS-related fatigue, dlPFC exhibited reduced resting state functional connectivity with some areas within parietal cortex and hyperconnectivity with others. This contrasts with a recent systematic review [18] which concluded that structural atrophy of thalamus and prefrontal cortex and greater functional connectivity in premotor and supplementary motor cortices underlies fatigue in MS. Notably, most patients described in the included studies experienced substantial disability. It is important to appreciate that feelings of mental/cognitive fatigue appear early in MS, often before overt signs of disability [40], a time when there is likely still capacity for neuroplasticity and patients could potentially benefit from restorative approaches such as neuromodulation. Despite lack of clarity regarding fatigue neuromodulation targets, several groups have proceeded to clinical trials [12]. For instance, transcranial direct current stimulation (tDCS) applied over the left dlPFC combined with cognitive training reduced feelings of fatigue in MS patients [15]. Using a different target, tDCS applied over right parietal cortex reduced cognitive fatigability in a vigilance task, but not subjective fatigue [13]. In a pilot study comparing tDCS applied to either left dlPFC or right posterior parietal cortex, targeting left dlPFC proved most effective at ameliorating MS fatigue [14]. Our results point to dlPFC, and more so left dlPFC, as a potential neuromodulation target. However left dlPFC demonstrated reduced hemodynamic coherence during cognitively fatiguing tasks rather than a more straightforward increase or decrease in activation, suggesting inconsistencies in intra-regional synaptic function and/or neuro-vascular coupling. It would seem reasonable to pair neuromodulation with targeted cognitive training in order to synchronize neuronal pools in the region and potentially stimulate neuroplasticity for longer lasting benefits.

## Intraregional disruption of hemodynamic response and neurovascular coupling

During single and dual tasks, healthy control subjects exhibited consistent hemodynamic responses among channels of the PFC. This synchrony was disrupted during cognitive tasks in the MS group particularly in bilateral dlPFC. While others have reported fatigue-induced disruption of connectivity between cortical and subcortical areas [39, 41–43], we show, for the first time using fNIRS, that asynchrony occurs within neuronal pools of the same brain region (PFC). Even though our sample of people with MS had no cognitive or physical disability, subclinical MS-related inflammation within the brain can disrupt the blood-brain-barrier and the endothelium of cerebral vessels, compromising neurovascular coupling [44]. Previous research supports that older persons [45] and persons who have low cardiorespiratory fitness [46] (which is common in MS [25]) also have compromised neurovascular coupling. It is possible that disorganized and inefficient activation of neuronal pools, due to impairments in neurovascular coupling during cognitive tasks, contributes to both mental fatigue and cortical activation asynchrony in MS. Such asynchrony may explain conflicting results of the hemodynamic response from other studies during fatiguing tasks among people with varying severities of MS symptoms; some groups report increased hemodynamic response [17] while other report suppressed response [47]. The region may respond differently depending on structural damage. Our previous work using transcranial magnetic stimulation [28], suggests that early in MS among persons with low disability (such as those recruited in the current study) there may be excessive excitability (glutamatergic) that progresses toward inhibition of the circuits over time. It is possible that the balance between excitation and inhibition may also

be disrupted in dlPFC neuronal pools impairing a synchronous hemodynamic response. Future research combining fNIRS and electroencephalogram or using novel statistical methods to describe both spatial and temporal parameters [48] may help elucidate these complex cortical responses.

## A paradoxical decrease in HbO during cognitive tasks

We witnessed a general decrease (rather than increase/peak) in HbO in PFC during the 20s task in healthy controls. The principle of neurovascular coupling describes that metabolic demand triggers an increase in oxygenated hemoglobin, which is represented by a peak. Using functional magnetic resonance imaging, decreased hemodynamic response is a recognized phenomenon referred to as negative BOLD (blood oxygen level dependent) response, sometimes referred to as 'task-induced deactivation' [49, 50]. This paradoxical negative response continues to be debated, but some hypotheses have been proposed. One explanation proposes that the region experiences 'vascular steal', which happens when interconnected areas 'steal' oxygenated blood from the region of interest, causing a steep decrease in blood supply in the latter [49, 51]. Because there is evidence for the role of subcortical structures and networks in MS-related fatigue [52–54], the deactivation of dlPFC may be part of the regulation of the subcortical networks connected to it. Another possible explanation could the deactivation of the default mode network (DMN). The DMN [55] comprises cortical and subcortical regions, including the dlPFC. While at rest or during mind wandering, the DMN is 'switched on', showing increased activation. However, when performing tasks in which sustained focus and attention is required, the DMN activity is suppressed to help minimize distractions from the environment. The function of the DMN is disrupted in many neurological disorders, including MS. A recent fMRI study investigating associations between DMN connectivity, fatigue and depression in multiple sclerosis showed higher DMN activity in fatigued subjects, even though they had low depression scores [56]. Another study of dynamic functional connectivity in MS-related fatigue showed lower basal ganglia-DMN connectivity with higher fatigue levels [41]. Because the DMN is supposed to deactivate during attention-demanding mental tasks, its increased activity reflects a failure to sustain attention and ignore distractions in people with fatigue. In our study, the prefrontal deactivation pattern is consistent between channels in healthy controls and disrupted in MS participants (Fig 4). Therefore, it is possible that the asynchrony seen in the MS group could be related to an inability to regulate the DMN during challenging cognitive tasks.

## Limitations

Although we screened participants in order to exclude those experiencing physical, cognitive and depressive symptoms, the MS group had significantly lower scores on the MoCA by one point, on average. The effects of even mild subclinical cognitive impairment could affect task-related changes in HbO. Due to capacity limitations within the fNIRS cap montage, we were not able to measure HbO in parietal cortices (simultaneously with dlPFC and frontopolar areas), regions that have been identified as being linked to fatigue and serving as potential neuromodulation targets [13, 14]. This limitation precluded our ability to compare hemodynamic responses in this region to dlPFC. We also chose to measure, as a first step, cognitive fatigue using subjective measures, recognizing that future research should consider the use of objective tools such as a computerized vigilance task.

## Conclusions

We show that in MS, dlPFC demonstrated disrupted hemodynamic coherence during both single and dual tasks, as evidenced by non-significant and negative correlations between fNIRS channels. In MS, reduced coherence occurred in left dorsolateral PFC during the single task but occurred bilaterally as the task became more challenging. Our findings point to dlPFC, but not frontopolar areas, as a potential target for neuromodulation to treat cognitive fatigue.

## Supporting information

**S1 Dataset.**
(XLSX)

## Author Contributions

**Conceptualization:** Bruna D. Baldasso, Michelle Ploughman.

**Data curation:** Bruna D. Baldasso, Syed Z. Raza, Sadman S. Islam, Isabella B. Burry, Caitlin J. Newell, Sydney R. Hillier, Michelle Ploughman.

**Formal analysis:** Bruna D. Baldasso, Syed Z. Raza, Sadman S. Islam, Michelle Ploughman.

**Funding acquisition:** Michelle Ploughman.

**Investigation:** Bruna D. Baldasso, Syed Z. Raza, Sadman S. Islam, Isabella B. Burry, Caitlin J. Newell, Sydney R. Hillier.

**Methodology:** Bruna D. Baldasso, Syed Z. Raza, Sadman S. Islam, Isabella B. Burry, Caitlin J. Newell, Sydney R. Hillier, Michelle Ploughman.

**Project administration:** Bruna D. Baldasso, Michelle Ploughman.

**Resources:** Michelle Ploughman.

**Supervision:** Michelle Ploughman.

**Validation:** Bruna D. Baldasso, Michelle Ploughman.

**Visualization:** Bruna D. Baldasso, Michelle Ploughman.

**Writing – original draft:** Bruna D. Baldasso, Michelle Ploughman.

**Writing – review & editing:** Bruna D. Baldasso, Michelle Ploughman.

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
