## [Decision Letter · Decision Letter 0]

29 Feb 2024

PONE-D-23-42776Disrupted hemodynamic response within dorsolateral prefrontal cortex during cognitive tasks among people with multiple sclerosis-related fatiguePLOS ONE

Dear Dr. Ploughman,

Thank you for submitting your manuscript to PLOS ONE. After careful consideration, we feel that it has merit but does not fully meet PLOS ONE’s publication criteria as it currently stands. Therefore, we invite you to submit a revised version of the manuscript that addresses the points raised during the review process.

We look forward to receiving your revised manuscript.

Kind regards,

Umer Asgher, PhD

Academic Editor

PLOS ONE

Journal Requirements:

**Additional Editor Comments:**

Based on the reviewers' feedback, comments and recommendations: Minor revision is recommended (details comments appended below).

Reviewers' comments:

Reviewer's Responses to Questions

**Comments to the Author**

1. Is the manuscript technically sound, and do the data support the conclusions?

Reviewer #1: Yes

Reviewer #2: Yes

2. Has the statistical analysis been performed appropriately and rigorously? 

Reviewer #1: Yes

Reviewer #2: Yes

3. Have the authors made all data underlying the findings in their manuscript fully available?

Reviewer #1: No

Reviewer #2: Yes

4. Is the manuscript presented in an intelligible fashion and written in standard English?

Reviewer #1: Yes

Reviewer #2: Yes

5. Review Comments to the Author

Reviewer #1: After reading the manuscript entitled ‘Disrupted hemodynamic response within dorsolateral prefrontal cortex during cognitive tasks among people with multiple sclerosis-related fatigue’ I find that on the whole, the authors have demonstrated rigor in the methodology and presentation of their article. However, I have identified a few minor concerns that deserve the authors' attention so that they can improve their manuscript.

- My first concern is the introduction. Although I find it precise and concise, in some places I think the authors could do a better job of referencing their words. For example, (lines 101-105) : ‘Functional near-infrared spectroscopy is … … … … (e.g. metal implants)’ I therefore suggest that authors propose references in this part of their manuscript.

- A second point that caught my eye was that in the method, specifically in the 'Fatigue Measures' section, line 183, the authors specified that they used two subjective tasks to measure fatigue. My question is, firstly, why they made this choice and, how they ensured the veracity of the participants' responses. Secondly, I'd like to know whether there are any objective measures of fatigue for this population.

- If I've understood correctly, the authors have focused solely on oxyhemoglobin for their hemodynamic treatment. But the hemodynamic response does not only concern oxyhemoglobin, but also deoxyhemoglobin. Even though we now know that HbO is more sensitive than HHB, recommendations in the literature today suggest at least presenting data from both signals. I strongly recommend that authors present the raw HHB data in supplementary material.

- Although I fully understand that the variable of interest in this study is the hemodynamic response, I'm still a little disappointed that the authors haven't given any indication of the behavioral results of the single and double tasks. So I'm curious to see what the results are.

- Lines 355, 366 and 369 : What's the difference between correlograms, correlelograms and correleograms? If this is a mistake, I suggest that the authors reread all the annotations underneath the figures and tables.

Reviewer #2: Manuscript Number: PONE-D-23-42776

Title: Disrupted hemodynamic response within dorsolateral prefrontal cortex during cognitive tasks among people with multiple sclerosis-related fatigue

Authors: Baldasso et al.

In this paper, the Authors study cortical hemodynamic responses in dorsolateral prefrontal cortex and frontopolar areas during single and dual cognitive tasks in persons with mental fatigue related to multiple sclerosis. In particular, they selected a group of subjects who were not affected by confounding conditions, such as depression, physical or cognitive impairments. The brain hemodynamic during the tasks was assessed by functional near infrared spectroscopy (fNIRS). The aim of the work was to identify brain areas that can be appropriate targets for the neurostimulation, an emerging treatment for cognitive fatigue.

The paper is well written and well organized. The scientific message of the paper is clear and supported by an appropriate literature, a well-designed experiment, and a comprehensive statistical analysis. Moreover, in this study it was employed fNIRS, an optical technique suitable to assess non-invasively brain hemodynamic and quite new in studies involving mental fatigue.

For these reasons, I deserve the paper for publication, provided that some minor points are addressed.

Minor points:

1) Page 16, Table 2.

Please, clarify what is parameter “F” reported in Table 2.

2) Page 18, Line 424.

I think the Authors mean “Figure 4” instead the reported “Figure 2”.

3) Page 18, Line 426.

I think the Authors mean “Figure 5” instead the reported “Figure 3”.

4) Page 18, Line 424.

I think the Authors mean “Figure 6” instead the reported “Figure 4”.

6. PLOS authors have the option to publish the peer review history of their article (what does this mean?). If published, this will include your full peer review and any attached files.

Reviewer #1: No

Reviewer #2: No

---

## [Author Response · Author response to Decision Letter 0]

27 Mar 2024

Reviewer #1: After reading the manuscript entitled ‘Disrupted hemodynamic response within dorsolateral prefrontal cortex during cognitive tasks among people with multiple sclerosis-related fatigue’ I find that on the whole, the authors have demonstrated rigor in the methodology and presentation of their article. However, I have identified a few minor concerns that deserve the authors' attention so that they can improve their manuscript.

R1.1 - My first concern is the introduction. Although I find it precise and concise, in some places I think the authors could do a better job of referencing their words. For example, (lines 101-105) : ‘Functional near-infrared spectroscopy is … … … … (e.g. metal implants)’ I therefore suggest that authors propose references in this part of their manuscript.

This is a very good point. We have now added two references that provide some perspective and review of the field (Chen 2020 and Pinti 2020).

R1.2 - A second point that caught my eye was that in the method, specifically in the 'Fatigue Measures' section, line 183, the authors specified that they used two subjective tasks to measure fatigue. My question is, firstly, why they made this choice and, how they ensured the veracity of the participants' responses. Secondly, I'd like to know whether there are any objective measures of fatigue for this population.

The reviewer is correct in that we did not measure cognitive fatigue objectively. We chose the two subjective measures, which are both participant-reported, based on our previous research (new references added #25 and 29 for the State Fatigue VAS which represents immediate feelings of fatigue) and previous research validating the tool (FSMC representing fatigue “you feel in normal day-to-day life”). It is difficult to assess the veracity of the responses but we can only assume that these represent the persons own experience at that time.

We could have used performance on a computerized vigilance task but then this would become both the intervention (programmed to provide both single and dual talk challenges) and the measurement. This is an area for future research.

We have added this note to the Limitations section “We also chose to measure, as a first step, cognitive fatigue using subjective measures, recognizing that future research should consider the use of objective tools such as a computerized vigilance task.”

R1.3- If I've understood correctly, the authors have focused solely on oxyhemoglobin for their hemodynamic treatment. But the hemodynamic response does not only concern oxyhemoglobin, but also deoxyhemoglobin. Even though we now know that HbO is more sensitive than HHB, recommendations in the literature today suggest at least presenting data from both signals. I strongly recommend that authors present the raw HHB data in supplementary material.

The reviewer is absolutely correct. We have now provided an Excel file with both the HbO (oxyhemoglobin) and HbR (de-oxyhemoglobin) values for each brain area of interest for every subject in the Supplementary File. This is mentioned now in the text line 263-264.

R1.4- Although I fully understand that the variable of interest in this study is the hemodynamic response, I'm still a little disappointed that the authors haven't given any indication of the behavioral results of the single and double tasks. So I'm curious to see what the results are.

This is a very good point. We asked participants to provide their answers to the single and dual tasks audibly. On hindsight, we could have audio recorded the answers and provided a metric of response errors or delays. We unfortunately do not have those results at all.

R1.5- Lines 355, 366 and 369 : What's the difference between correlograms, correlelograms and correleograms? If this is a mistake, I suggest that the authors reread all the annotations underneath the figures and tables.

Sorry for this oversight. The correct spelling is ‘Correlogram’ which we have corrected throughout.

Reviewer #2: In this paper, the Authors study cortical hemodynamic responses in dorsolateral prefrontal cortex and frontopolar areas during single and dual cognitive tasks in persons with mental fatigue related to multiple sclerosis. In particular, they selected a group of subjects who were not affected by confounding conditions, such as depression, physical or cognitive impairments. The brain hemodynamic during the tasks was assessed by functional near infrared spectroscopy (fNIRS). The aim of the work was to identify brain areas that can be appropriate targets for the neurostimulation, an emerging treatment for cognitive fatigue.

The paper is well written and well organized. The scientific message of the paper is clear and supported by an appropriate literature, a well-designed experiment, and a comprehensive statistical analysis. Moreover, in this study it was employed fNIRS, an optical technique suitable to assess non-invasively brain hemodynamic and quite new in studies involving mental fatigue.

For these reasons, I deserve the paper for publication, provided that some minor points are addressed.

Minor points:

R2.1 1) Page 16, Table 2.

Please, clarify what is parameter “F” reported in Table 2.

Thank you for pointing out this oversight. We have now added the F-value explanation to the Statistical Analysis section line 289 “Peak HbO, time-to-peak and functional connectivity scores were then compared between groups and conditions (single and dual task) using a factorial ANOVA (group X condition), reporting the variation between sample means as F-value.”

R2.2 2) Page 18, Line 424.

I think the Authors mean “Figure 4” instead the reported “Figure 2”.

Thank you for noticing these errors which have since been corrected.

R2.3 3) Page 18, Line 426.

I think the Authors mean “Figure 5” instead the reported “Figure 3”.

Thank you for noticing these errors which have since been corrected.

R2.4 4) Page 18, Line 424.

I think the Authors mean “Figure 6” instead the reported “Figure 4”.

Thank you for noticing these errors which have since been corrected.

---

## [Decision Letter · Decision Letter 1]

22 Apr 2024

Disrupted hemodynamic response within dorsolateral prefrontal cortex during cognitive tasks among people with multiple sclerosis-related fatigue

PONE-D-23-42776R1

Dear Dr. Ploughman,

We’re pleased to inform you that your manuscript has been judged scientifically suitable for publication and will be formally accepted for publication once it meets all outstanding technical requirements.

Kind regards,

Umer Asgher, PhD

Academic Editor

PLOS ONE

Additional Editor Comments (optional):

Reviewers' comments:

Reviewer's Responses to Questions

**Comments to the Author**

1. If the authors have adequately addressed your comments raised in a previous round of review and you feel that this manuscript is now acceptable for publication, you may indicate that here to bypass the “Comments to the Author” section, enter your conflict of interest statement in the “Confidential to Editor” section, and submit your "Accept" recommendation.

Reviewer #1: All comments have been addressed

Reviewer #2: All comments have been addressed

2. Is the manuscript technically sound, and do the data support the conclusions?

Reviewer #1: Yes

Reviewer #2: (No Response)

3. Has the statistical analysis been performed appropriately and rigorously? 

Reviewer #1: Yes

Reviewer #2: (No Response)

4. Have the authors made all data underlying the findings in their manuscript fully available?

Reviewer #1: Yes

Reviewer #2: (No Response)

5. Is the manuscript presented in an intelligible fashion and written in standard English?

Reviewer #1: Yes

Reviewer #2: (No Response)

6. Review Comments to the Author

Reviewer #1: I'm satisfied with the various corrections made by the authors, and in my opinion, the work is ready for publication.

Reviewer #2: (No Response)

7. PLOS authors have the option to publish the peer review history of their article (what does this mean?). If published, this will include your full peer review and any attached files.

Reviewer #1: No

Reviewer #2: No

---

## [Editor Report · Acceptance letter]

27 May 2024

PONE-D-23-42776R1 

PLOS ONE

Dear Dr. Ploughman, 

I'm pleased to inform you that your manuscript has been deemed suitable for publication in PLOS ONE. Congratulations! Your manuscript is now being handed over to our production team.

Kind regards, 

on behalf of

Dr. Umer Asgher 

Academic Editor

PLOS ONE